# Three-Dimensional Analysis of Alveolar Bone Morphological Characteristics in Skeletal Class II Open Bite Malocclusion: A Cone-Beam Computed Tomography Study

**DOI:** 10.3390/diagnostics13010039

**Published:** 2022-12-23

**Authors:** Yingying Tang, Jingfeng Xu, Yun Hu, Yumei Huang, Yang Liu, Baraa Daraqel, Leilei Zheng

**Affiliations:** 1Department of Stomatological Hospital of Chongqing Medical University, Chongqing Key Laboratory of Oral Diseases and Biomedical Sciences, Chongqing Municipal Key Laboratory of Oral Biomedical Engineering of Higher Education, Chongqing 401147, China; 2Department of Stomatological Hospital of Chongqing Medical University, Chongqing 400016, China

**Keywords:** orthodontics, 3D diagnostics, alveolar bone morphology, class II open bite, CBCT

## Abstract

This cross-sectional research evaluated the morphological characteristics of alveolar bone in skeletal class II open-bite individuals compared to skeletal class II and class I non-open-bite individuals. A sample of 82 patients (all were in cervical vertebral stage 6) was divided into three groups (class II OB group, n = 29; class II NOB group, n = 29; class I NOB group, n = 24) according to bilateral molar relationship, ANB angle, and anterior overbite measured by cephalometric analysis. The evaluation was performed with specialized software (Mimics 21.0) and initial cone-beam computed tomography (CBCT) data. The alveolar bone height and thickness of the whole dentition area and alveolar crest level and root apex position of the incisors were measured with a series of appropriate CBCT images. One-way analysis of variance followed by the Tukey post hoc test and the Kruskall–Wallis test were performed for statistical comparisons. The class II open-bite group had increased alveolar bone height for the maxillary first molar and decreased alveolar bone height for mandibular molars compared to the class II non-open-bite group (*p* < 0.05 for both). Furthermore, there were significant negative correlations between the alveolar bone heights of the upper first and second molars (A6-height, A7-height) and overbite (both *p* < 0.01). The alveolar thicknesses of all measured teeth were generally reduced in the class II OB group.

## 1. Introduction

Anterior open bite is defined as a lack of contact between the incisal edges of maxillary and mandibular anterior teeth in centric relation [1]. Anterior open-bite malocclusion is characterized by complex etiology, limited treatment, and a strong tendency for relapse [1,2,3,4]. Skeletal class II open bite is characterized by excessive vertical growth of the upper jaw, shortening of the ramus of the lower jaw, or a combination of both [2]. Thus, class II open bite with vertical and sagittal anomalies is challenging for orthodontists [5]. Many previous studies have indicated that skeletal class II open bite featured the mandible in a backward and downward position, a larger mandibular plane angle, and increased anterior facial height [6,7].

To achieve closure of anterior teeth, traditional orthodontic camouflage treatment for anterior open bite involves intruding the molars or extruding the incisors [8]. Extrusion of incisors is still used, although many studies have shown the inappropriateness of extrusion of incisors. Numerous case reports have demonstrated that intrusion of molars utilizing the skeletal anchorage system to treat open bite can achieve good therapeutic outcomes and stability similar to surgical approaches [8,9,10,11,12]. However, there are controversial results for the dentoalveolar morphology of open-bite malocclusion. Several studies found that the dentoalveolar heights of the maxillomandibular molars both increased more in class II open bite than in the control group [7,13]. Conversely, others reported that only the posterior maxillary dentoalveolar height increased, and there was no difference in anterior dentoalveolar heights [14]. Some studies reported that posterior and anterior dentoalveolar heights increased in the open-bite group [1,15,16]. In addition, Nahoum even found that mandibular dentoalveolar height decreased in an open-bite patient [6]. The factors contributing to the contradictory results may have been the different inclusion criteria (such as ethnicity, gender, and sagittal skeletal features), the difference in experimental methods, and the various statistical analyses used.

It is known that the extension of the maxillary sinus affects the maxillary posterior alveolar bone and the roots of the upper posterior teeth, especially during bodily movement or the intrusion of the teeth involved. The relationship between the mandibular canal and the roots is similarly essential [17]. Most previous studies have used cephalometric analyses to assess the dentoalveolar morphology of incisors and first molars [18,19,20,21]. Supontep Teerakanok et al. found that routine lateral cephalograms were not suitable for evaluation of dentoalveolar thicknesses and heights [22]. CBCT has the advantages of high spatial resolution and of not involving magnification, allowing it to reflect the three-dimensional spatial structure of the maxillofacial tissue objectively [17,23]. Werinpimol Kosumarl et al. [24] reported there were no differences in the distances from the root apices of posterior teeth to the maxillary sinus and mandibular canal in patients with skeletal open bite compared to the normal bite group.

In orthodontic camouflage treatment of class II open bite, extensive sagittal and vertical movements of the anterior teeth are often required to improve the profile and achieve normal overbite/overjet. The alveolar bone morphology and periodontal tissue of incisors set limits on the effects of treatment. There are few studies on alveolar bone thickness in skeletal class II open bite.

Thus, the purpose of this study was to accurately and comprehensively evaluate the dentoalveolar morphology of the entire dentition in skeletal class II open-bite individuals compared to class I/II non-open-bite individuals. Furthermore, we aimed to identify the dentoalveolar factors associated with the anterior overbite.

## 2. Materials and Methods

### 2.1. Selection and Grouping of Sample

In this retrospective research, 82 patients’ pretreatment lateral cephalograms and cone-beam computed tomography (CBCT) scans were selected from the database of the Orthodontics Department at the Stomatological Hospital archives of Chongqing Medical University according to the inclusion criteria. In this department, records are taken to evaluate temporomandibular joints, supernumerary and impacted teeth, root resorption and fractures, implant sites, airways, maxillofacial tumors and cysts, craniofacial deformities, and syndromes. The following are the inclusion criteria that were used for selecting samples: (1) complete eruption of second molars and cervical vertebral stage 6; (2) complete dentition, without extra or missing teeth (except third molars); (3) no crowding or less than 4 mm with single permanent dentition; (4) no anterior or posterior crossbite; (5) no history of previous orthodontic or prosthodontic treatment; (6) no evidence of cleft lip and palate history, congenital syndrome, facial or dental trauma, or craniofacial anomalies; (7) symmetrical face and chin consistent with the midplane line; (8) roots of the posterior maxillary teeth below the maxillary sinus floor according to the panoramic image derived from the CBCT. As a result, the sample was composed of 82 patients (19 men, 63 women) ranging in age from 13 to 35 years old.

Samples were divided into three groups according to the degree of anterior overbite, the sagittal skeletal relation, and the bilateral molar relationship. In addition, overbite and ANB angle were both confirmed by cephalogram. The sample groups were as follows: group 1 (class II, open bite, n = 29)—anterior overbite < 0 mm, ANB > 5°, bilateral molar class II relationship; group 2 (class II, non-open bite, n = 29)—overbite > 1 mm, ANB > 5°, bilateral molar class II relationship; group 3 (class I, normal overbite and overjet, n = 24)—1 mm < overbite < 4 mm, 0.5 mm < overjet < 3 mm), 1° ≤ ANB ≤ 4°, bilateral molar class I relationship. Class II malocclusions were defined as bilateral molar apical to apical or completely distal.

### 2.2. Acquisition of Imaging Data and Maxillofacial 3D Reconstruction

The same digital X-ray cephalometric machine (ProMax; Planmeca, Helsinki, Finland) was used for scanning with all subjects, with a voltage of 80 KV, current time of 10 mA, and exposure time of 0.5 s. All CBCT images were obtained and scanned with the same CBCT scanner (KaVo Dental GmbH, Bismarckring, Germany) with a current time of 5 mA, voltage of 120 KV, exposure time of 8.9 s, and resolution of 0.4 mm. During the two scans, patients were required to maintain a natural head position and bite their teeth with intercuspation. The cervical vertebral maturation stages were visually assessed on the lateral cephalograms by one investigator and confirmed by a second. Combined with Jarabak analysis, 14 angular and 2 linear ratio measurements were selected to determine the dentoskeletal morphologies of maxillary and mandibular molars (Figure 1).

All CBCT image data were imported into Mimics 21.0 software (Materialise, Leuven, Belgium) in the DICOM format for maxillofacial 3D reconstruction and measurement. The subspinale point (point A), supramental point (point B), and the center point of the clinical crown (point FA) of the upper- and lower-jaw anterior teeth are marked on the 3D model. The midpoint of the clinical crown center point of the bilateral maxillary central incisor is denoted as point #1121FA, and the mandibular midpoint is also marked. The maxillary or mandibular anterior reference plane (UARP or LARP), and the maxillary or mandibular anterior basal bone plane (UABP or LABP) were also determined (Figure 2).

The mandibular anterior reference plane (LARP) is the plane passing through the FA point #3343 and the center of the #31.41 FA point; the maxillary anterior basal bone plane (UABP) is the plane passing through the A point parallel to the maxillary anterior reference plane (UARP); the mandibular anterior basal bone plane (LABP) is the plane passing through the B point parallel to the mandibular anterior reference plane (LARP).

### 2.3. Measurement Methods and Items

The 3D view was adjusted to obtain the right plane for accurate alveolar bone measurements. The midsagittal view of the teeth was selected as the measurement plane (Figure 3).

The alveolar heights and thicknesses of the anterior and posterior teeth at three levels were evaluated. The mesial and distal roots of the mandibular molars were measured separately, and the average value was taken to represent the alveolar bone height of the molars. Moreover, the distance from the labial or lingual crest to the CEJ (recorded as UBH or LBH) and the distance from the apex to the basal bone plane (line OH) for the anterior teeth were taken into consideration (Figure 4, Figure 5 and Figure 6).

### 2.4. Statistical Analysis

All statistical analyses were completed using SPSS software for Windows (version 26; IBM SPSS, Chicago, III). Statistical significance was set at *p* < 0.05 for all tests. The Chi-squared test was used to analyze differences in gender composition among the three groups. The normality and variance in the homogeneity of the experimental data were confirmed using the Shapiro–Wilk test and the Levene test, respectively. The above test results were used to determine whether the data required further parametric or non-parametric testing. One-way analysis of variance followed by the Tukey post hoc test and the Kruskal–Wallis test were used for further statistical evaluations. Relationships between the overbite and the alveolar parameters were evaluated using the Pearson correlation test and Spearman correlation test.

All the tracings and measurements were undertaken by one examiner. Twenty randomly selected subjects were re-measured two weeks later to evaluate intraoperator reliability, and the intraclass correlation coefficient (ICC) values ranged from 0.81 to 0.95.

The sample size was calculated by considering a mean difference of 1.26 mm as a clinically relevant difference between groups with a standard deviation of 2.5 mm (obtained from a preliminary pilot study in which the means of the U6-height differences between class II open-bite patients and controls were compared). Power analysis showed that each group of at least 22 samples achieved a statistical power of 80% at a one-sided significance level of 0.05.

## 3. Results

### 3.1. Sample

From the elemental composition of the sample, it can be seen that there were no significant differences in gender distribution or age in each group (Table 1).

### 3.2. Comparative Description of Cranial–Maxillary Relationships

Group 1 (class II, OB) and group 2 (class II, NOB) showed similar retrognathic positions or mandibular hypoplasia. Furthermore, group 1 (class II, OB) had significantly increased mandibular plane angles (SN-GoGn, FMA), Jarabak’s sums, gonial angles (Ar-Go-Me), lower gonial angles (N-Go-Me), and upper face height to total facial height ratios (ANS-ME/N-Me), as well as decreased posterior–anterior ratios (S-Go/N-Me), than non-open-bite groups (all *p* < 0.01). No differences were observed in saddle angle (N-S-Ar) or articular angle (S-Ar-Go) between groups (*p* > 0.05) (Table 2).

### 3.3. Alveolar Features of Central Incisors

No significant differences were found for U1-SN (°), the alveolar height of the upper right central incisor (A1-height), or A1-OH, but the buccal alveolar bone attachment level (A1-UBH) in group 1 (class II, OB) was significantly larger than that in the other two groups (*p* < 0.05). The palatal alveolar bone attachment (A1-LBH) in class II groups was higher than that in group 3 (class I, NOB, *p* < 0.01). In addition, the alveolar thickness of A1 (A1-thickness) in group 1 was significantly decreased compared to the other two groups (*p* < 0.01). The mandibular incisor in group 2 (class II, NOB) was significantly more proclined (IMPA) compared to the other groups (*p* < 0.01). C1-UBH in group 1 (class II, OB) was significantly higher than in group 3 (class I, NOB, *p* < 0.01). The alveolar thickness of C1 (C1-thickness) in group 1 was significantly decreased compared to the other two groups (*p* < 0.001) (Table 3).

### 3.4. Alveolar Bone Morphological Characteristics

The alveolar bone height of the maxillary canine (A3-height) in group 1 (class II, OB) was lower than in group 3 (class I, NOB, *p* < 0.05), and C3-height in group 1 was lower than in group 2 (class II, NOB, *p* < 0.05); furthermore, the maxillomandibular canine had the lowest alveolar bone thickness (*p* < 0.001). There was no significant difference in the alveolar bone heights of the maxillomandibular second premolars (A5-height, C5-height) between the three groups, but the alveolar bone thickness in group 1 was lower than in group 3 (both *p* < 0.01). The alveolar bone height of the upper first molar (A6-height) in group 1 was significantly increased compared to group 2 (*p* < 0.05), and it was numerically larger than in group 3. No significant difference was found in the alveolar bone heights of the upper second molars (A7-height). The average alveolar bone height of the lower first molar and second molar (C6-height, C7-height) was significantly decreased in group 1 compared to group 2 (*p* < 0.05). The alveolar bone thickness of C6M, C6D, and C7M in group 1 was the lowest among the three groups (*p* < 0.01). The alveolar thicknesses of the maxillary molars (A6-thickness, A7-thickness) and the distal root of the mandibular second molar (C7D-thickness) in group 1 were both smaller than those in group 2 (*p* < 0.05) (Table 4).

### 3.5. Correlation Analysis of Anterior Overbite

In the class II open-bite group, there were only significant negative correlations between the alveolar bone heights of the upper first molar and second molar (A6-height, A7-height) and overbite (both *p* < 0.01). In addition, the alveolar bone height and thickness of the upper incisor (A1-height, *p* < 0.01; A1-thickness, *p* < 0.05) and the alveolar bone height of the upper canine (A3-height, *p* < 0.05) were significantly positively correlated with overbite. The distance from the root apex of the upper incisor to the basal bone plane (A1-OH) was significantly positively correlated with overbite only in group 1 (*p* < 0.01) (Table 5).

## 4. Discussion

In this study, the alveolar morphology of the entire dentition area in class II open-bite patients was accurately evaluated by CBCT using the inherent anatomical structure as the boundary. The root positions and tooth inclinations of the incisors are described in detail.

Except for individuals with severe skeletal deformities, the treatment of open bite at adolescence mainly involves non-surgical orthodontic camouflage treatment. Therefore, although vertical maxillofacial development continues into adulthood, there was no strict age limit in the present study. The cervical vertebral stage and the eruption of second molars were selected as the standard for maturation growth.

The SNA value represents the skeletal position of the maxilla, and it can be concluded from results that the maxilla was less prognathic in group 1 (class II OB) than in group 2 (class II NOB). This contradicts other studies reporting similar maxillae in class II OB and NOB groups. With regard to the skeletal position of the mandible (SNB) in skeletal open bite, many previous investigations reported a significantly increased mandibular plane angle and more retrognathic mandible due to backward rotation of the mandible [7,15,25,26]. The results of this study are consistent with the above conclusions. The SNB value in group 1 was significantly lower than in other groups, and the S-Ar-Go, Ar-Go-Me, Jarabak’s sum, N-Go-Me, and lower anterior face height all significantly increased in the OB group.

Most previous studies utilized lateral cephalograms to evaluate central incisor height and molar height based on the mandibular plane and the palatal plane [13,19,20], which does not accurately and comprehensively reflect the alveolar morphology [22].

In 1925, Lundström first proposed the concept of the apical basal bone, defining it as the stable boundary between the alveolar bone and the maxilla or mandible [27]. In 2011, Kim first constructed the maxillary and mandibular basal bone planes passing through point A (subspinale) and point B (supramental) using 3D modeling and CBCT [28,29]. In the current study, considering the relatively downward-rotated mandible in class II open-bite patients, the anterior dentoalveolar morphology was evaluated with the basal bone plane as the reference, and the evaluation of the posterior teeth used the corresponding anatomical structure (maxillary sinus and inferior alveolar canal) as references [24,30].

Many studies on the dentoalveolar compensatory mechanism or alveolar bone morphology of skeletal open bite are not precisely consistent [13,19,20,21]. Josef Kucera et al. considered that skeletal open bite was compensated with incisor elongation and inclination [19]. Arriola-Guillén et al. found that upper incisor height in a skeletal open-bite class II group was greater than that in the class I control group [31]. In contrast, other studies reported that incisor height was lower than in the control group [6,25]. The current study did not find compensatory elongation of the upper and lower incisors in skeletal class II open-bite malocclusion except for the decline at the alveolar crest level. Asli Baysal reported that a class II high-angle subgroup had thinner spongious bone in the lower incisor compared to a class II average-angle subgroup [32]. In this study, the alveolar bone of the upper and lower incisors in the class II open-bite group was the thinnest among the three groups.

The inclination of incisors in skeletal open bite depends on the altered maxilla–mandibular relationship and the dental compensation [33]. The inclination of the upper incisor (U1-SN°) in the class II OB group did not differ from that in other groups. However, the proclination of the lower incisor (IMPA) in the class II OB group was less than that in the class II NOB group. These results are in agreement with a previous study that reported that the inclination of incisors in skeletal class II open bite was no different from the control group (class I) [20]. The reduced level of the labial alveolar crest or thinner alveolar bone may have been the factors limiting the proclination of the lower incisor in the class II OB group. Ji Haining found that proclination of the incisor was significantly positively correlated with the alveolar bone attachment on the labial side of the upper and lower incisors in class II malocclusion [34].

Increased molar height is a common feature of skeletal open bite, but controversy remains as to whether the height of the upper and lower molars increases [19,35,36]. Isaacson et al. concluded that the height of the posterior maxillary alveolar process was the most critical factor in determining whether a high or low mandibular plane growth pattern occurs [35]. Some studies only found increased upper molar height in open-bite malocclusion [1,14,15,37]. However, others reported that the lower molar height in the skeletal open-bite group was larger than that in the control group [6,20,21]; even Nahoum considered that the dentoalveolar height of the mandibular first molar was smaller in the open-bite group than in the control group [6]. In the present study, there was no difference in the alveolar bone heights of the premolars among the three groups. Our results revealed that the alveolar bone height of the upper first molar was increased rather than that of the upper second molar in the open-bite group. Additionally, the alveolar bone heights of the lower first molar and lower second molar in the class II OB group decreased more than those in the class II NOB group. The increased alveolar height of the maxillary first molar may be one of the essential causes of skeletal class II open bite and may be the dominant factor in orthodontic compensatory treatment. The alveolar bone thickness of posterior teeth in class II open bite is generally reduced compared to other groups. Weaker alveolar bones across the whole dentition are the main manifestation of skeletal class II open bite.

Non-surgical orthodontic methods for treating open bite often involve the extrusion of anterior teeth and the intrusion of posterior teeth [38]. Studies regarding the multiloop edgewise archwire (MEAW) technique [39] and clear aligners [40] have reported the effect of the extrusion of anterior teeth. In the class II open-bite group, anterior overbite was positively correlated with the alveolar height and thickness of the upper incisor, the distance from the root apex of the upper incisor to the basal bone, and the alveolar height of the upper canine, so the alveolar morphology of the upper incisor had an important influence on anterior overbite. Although extrusion of anterior teeth is beneficial for access to incisal overlap, maxillary incisor extrusion is unstable and may compromise periodontal structures and smile esthetics [41]. Combined with our results showing that the alveolar bone height of the maxillary incisor did not increase in the open-bite group, the extrusion of incisors to obtain normal overbite is not recommended.

The intrusion of posterior teeth is considered to bring about the counterclockwise rotation of the mandible, thus increasing facial height and improving convex profiles [8,9]. Overbite in the class II open-bite group was negatively correlated with the alveolar bone height of the maxillary molars (A6-height, A7-height). In agreement with our results, many studies provide evidence supporting the intrusion of the maxillary molar to improve open bite, and this method has achieved good effects [5,42,43,44,45,46]. Man-Suk Baek et al. [44] reported that, in their study, the maxillary first molars were intruded by 2.39 mm on average and incisal overbite increased by a mean of 5.56 mm during treatment. Although this study found decreased alveolar bone height for the lower posterior teeth in the class II open-bite group, other studies were able to correct anterior open bite by intruding mandibular posterior teeth [47].

The thickness of alveolar bone was generally decreased in class II open-bite individuals compared to non-open-bite groups in our study, showing that it is necessary to pay attention to the range of tooth movement to prevent the occurrence of dehiscence and fenestration.

In light of the findings of this study, we recommend correcting class II open-bite malocclusion with appropriate maxillary first molar intrusion and avoiding extrusion of the incisors. At the same time, clinicians need to pay close attention to the position of the anterior teeth to ensure that the teeth are located in the alveolar bone and avoid the occurrence of dehiscence or fenestration.

Influenced by genetic factors, the incidence of anterior open bite varies greatly among different ethnicities, with the prevalence among the Caucasian population in America being 2.9% [2]. However, more comparative studies are needed to determine whether there are ethnic differences in the alveolar bone morphology of open-bite malocclusion.

The sample size of the class I NOB group was smaller than that of the other groups in the present study. Due to the strict inclusion criteria, it was relatively difficult to find proper individuals to construct group 3. Another limitation was that sex differences were not explored, which would require a larger sample size and further research.

## 5. Conclusions

The class II open-bite group had increased alveolar bone height for the maxillary first molar and decreased alveolar bone height for the mandibular molars compared to the class II non-open-bite group.

The mandibular incisor was less proclined in the class II open-bite group than in the class II non-open-bite group. In contrast, no significant differences in the inclination and alveolar bone height of the maxillary incisors were found among the three groups.

The thickness of alveolar bone was generally decreased in class II open-bite individuals compared to non-open-bite groups.

## Figures and Tables

**Figure 1 diagnostics-13-00039-f001:**
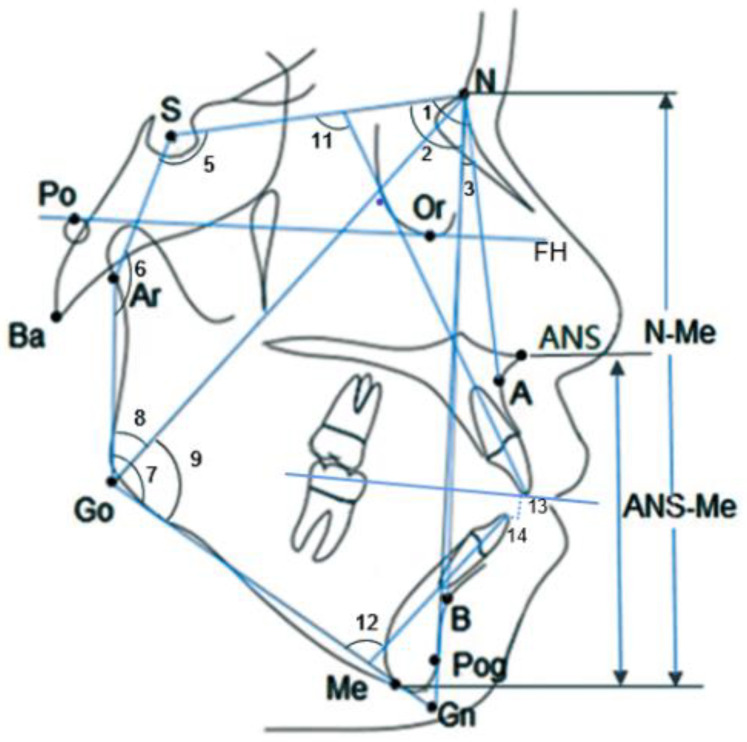
Angular and linear ratio measurements on lateral cephalograms. (1) SNA (°); (2) SNB (°); (3) ANB (°); (4) SN/GoGN (°); (5) N-S-Ar (°); (6) S-Ar-Go (°); (7) Ar-Go-Me (°); (8) Ar-Go-N (°); (9) N-Go-Me (°); (10) Jarabak‘s sum, the sum of angles N-S-Ar (°), S-Ar-Go (°), and Ar-Go-Me (°); (11) U1-SN (°); (12) L1-MP (°); (13) overbite (OB, mm); (14) overjet (OJ, mm); (15) S-Go/N-Me%; (16) ANS-Me/N-Me%.

**Figure 2 diagnostics-13-00039-f002:**
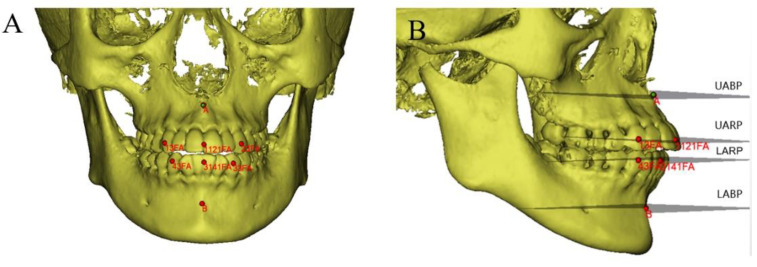
Three-dimensional craniomaxillofacial model and basal bone planes. (**A**) Craniomaxillofacial model and dental arch marking points; (**B**) the maxillary anterior reference plane (UARP) is the plane passing through the point #13.23 FA and the center of the #11.21 FA point.

**Figure 3 diagnostics-13-00039-f003:**
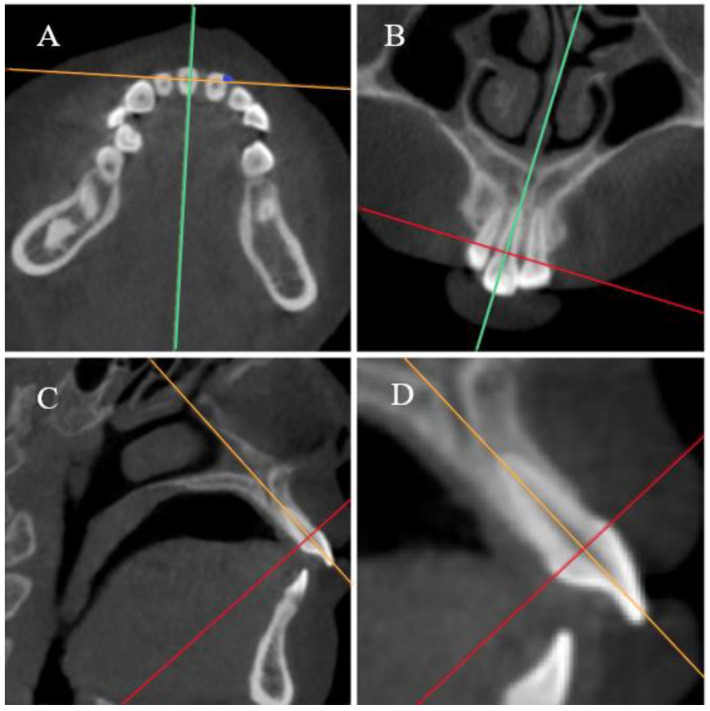
Location of the tooth measurement plane. (**A**–**C**) The axial plane, coronal plane, and sagittal plane of the teeth, respectively. Red, green, and orange lines in the pictures represent the axial plane, sagittal plane, and coronal plane, respectively. (**A**) The location of the axial plane was obtained by drawing the red line through the CEJ of the selected tooth in sagittal view; then, the orange line was rotated until the intersecting line was the shortest. (**B**) The green line was rotated until it passed through the root apex and the midpoint of the incisal margin. (**C**) The orange guideline was rotated until it passed through the root apex and the midpoint of the CEJ line. (**D**) A magnification of the sagittal view shows the measurement plane.

**Figure 4 diagnostics-13-00039-f004:**
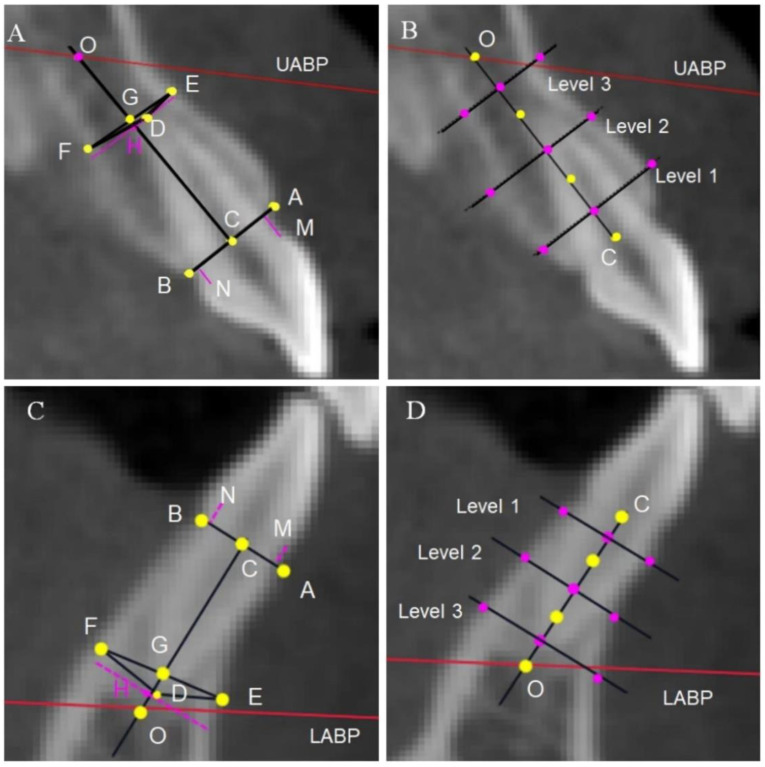
Landmarks and linear measurements of anterior tooth. (**A**–**D**) Measurements of the upper and lower front teeth, respectively. A, M, N: Labial CEJ and palatal CEJ; A, B: labial alveolar crest and palatal alveolar crest; C: midpoint of AB line. D: Root apex point. E: The point on the labial wall closest to the root apex point; F: the point on the palatal wall closest to the root apex point; G: midpoint of the EF line; O: the intersection of the CG line and UABP; the CO line represents the long axis of the alveolar bone. The length of the CO line is the alveolar height. H: The foot point of a line perpendicular to the CO line at point D. B: The long axis of the alveolar bone (CO line) is divided into thirds (shown by the yellow dots), which mark the midpoint of each third line as purple dots. Three lines at different levels are drawn perpendicular to the long axis of the alveolar bone and extend from the labial to the palatal cortical plate. The distance between the two plates represents the alveolar thickness at level 1 (coronal third), level 2 (middle third), and level 3 (apical third), respectively. The vertical distances from the M and N points to the vertical line of the AB line represent the level of buccal or palatal alveolar bone attachment of the anterior teeth, respectively, indicated by UBH or LBH. The length of the OH line represents the relative positional relationship between the root tip of the central tooth and the basal bone plane.

**Figure 5 diagnostics-13-00039-f005:**
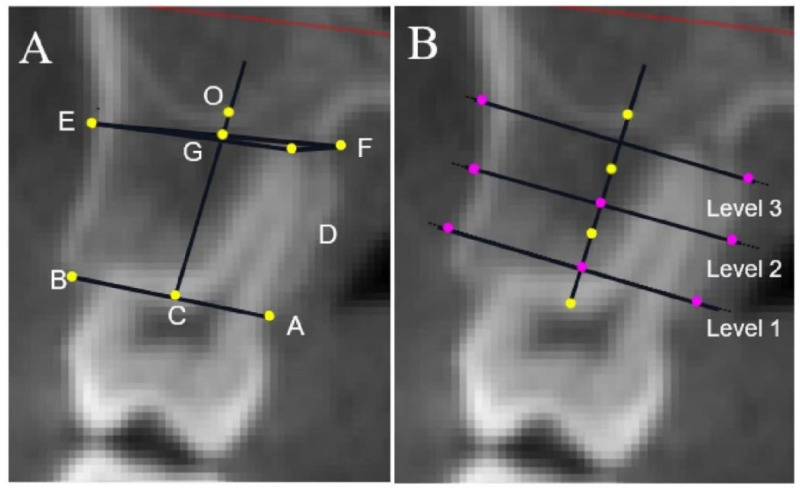
Landmarks and linear measurements of a maxillary posterior tooth. (**A**) A, B: Buccal alveolar crest and palatal alveolar crest; C: midpoint of AB line. D: Palatal root apex point. E: The point on the buccal wall closest to the root apex point; F: The point on the palatal wall closest to the root apex point; G: midpoint of the EF line; O: the intersection of the CG line and the floor of the maxillary sinus; the CO line represents the long axis of the alveolar bone. The length of the CO line is the alveolar height. (**B**) The long axis of the alveolar bone (CO line) is divided into thirds (shown by the yellow dots), which mark the midpoint of each third line as purple dots. Three lines at different levels are drawn perpendicular to the long axis of the alveolar bone and extend from the buccal to the palatal cortical plate. The distance between the two plates represents the alveolar thickness at level 1 (coronal third), level 2 (middle third), and level 3 (apical third), respectively.

**Figure 6 diagnostics-13-00039-f006:**
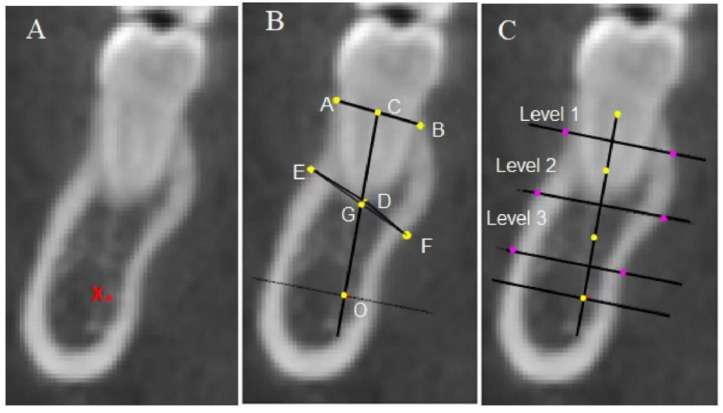
Landmarks and linear measurements of a mandibular posterior tooth. (**A**) X: Top point of the mandibular alveolar neural tube. (**B**) A, B: Buccal alveolar crest and lingual alveolar crest; C: midpoint of the AB line. D: Root apex point. E: The point on the buccal wall closest to the root apex point; F: the point on the lingual wall closest to the root apex point; G: midpoint of EF line; O: the foot point of a line perpendicular to the CG line at point X; the CO line represents the long axis of the alveolar bone. The length of the CO line is the alveolar height. (**C**) The long axis of the alveolar bone (CO line) is divided into thirds (shown by the yellow dots), which mark the midpoint of each third line as purple dots. Three lines at different levels are drawn perpendicular to the long axis of the alveolar bone and extend from the buccal to the lingual cortical plate. The distance between the two plates represents the alveolar thickness at level 1 (coronal third), level 2 (middle third), and level 3 (apical third), respectively.

**Table 1 diagnostics-13-00039-t001:** Summary of subject data.

	Group 1	Group 2	Group 3	*p* Value
Number	29	29	24	
Age (y; median (Q1~Q3))	18 (15.5~21.5)	20 (15~23)	21 (17.25~22)	NS ^b^
Gender (female/total)	22/29	20/29	21/24	0.278 ^c^

NS, not significant, ^b^ Kruskal–Wallis test, ^c^ Chi-squared test.

**Table 2 diagnostics-13-00039-t002:** Comparison of cephalometric measurements between three groups.

	Group 1 (Class II; OB)	Group 2 (Class II; NOB)	Group 3 (Class I; NOB)	*p* (Group)	Comparison Between Groups		
					1 vs. 2	1 vs. 3	2 vs. 3
SNA	80.43 ± 4.02	82.98 ± 2.76	81.11 ± 2.63	0.011 *	0.003 **	0.444	0.039 *
SNB	73.82 ± 3.85	76.3 ± 3.17	78.18 ± 2.31	0.000 †	0.004 **	0.000 †	0.038 *
ANB	6 (5.5~7.6)	6.4 (5.45~7.7)	3.2 (2.13~3.88)	0.000 †	1.000	0.000 †	0.000 †
SN-GoGn	43.3 (40.1~46.15)	35 (32.4~36.95)	33.5 (32.93~35.8)	0.000 †	0.000 †	0.000 †	1.000
OB	−2.79 ± 1.78	4.19 ± 1.4	2.67 ± 1.22	0.000 †	0.000 †	0.000 †	0.000 †
OJ	5.5 (3.9~7)	5 (3.8~6.95)	3.4 (2.25~3.75)	0.000 †	1.000	0.000 †	0.000 †
N-S-Ar	124.8 (121.2~127.5)	124.9 (122.4~126.55)	126.45 (125.23~129.78)	0.056	/	/	/
S-Ar-Go	153.3 (149~158.15)	150.9 (148.4~155.1)	146.95 (145~151.03)	0.007 **	1.000	0.007 **	0.056
Ar-Go-Me	126.96 ± 6.09	119.18 ± 4.83	119.11 ± 5.2	0.000 †	0.000 †	0.000 †	0.964
Jarabac’s sum	405.6 (401~408.9)	396.4 (393~398.4)	393.65 (392.45~396.3)	0.000 †	0.000 †	0.000 †	1.000
Ar-Go-N	44.42 ± 3.39	43.76 ± 2.88	44.23 ± 3.44	0.731	0.443	0.837	0.599
N-Go-Me	82.53 ± 4.47	75.4 ± 3.33	74.87 ± 3.1	0.000 †	0.000 †	0.000 †	0.607
S-Go/N-Me%	58.1 (56.5~62.45)	63.4 (61.85~64.9)	63.7 (62.2~64.88)	0.000 †	0.001 **	0.001 **	1.000
ANS-ME/N-Me%	56.25 ± 1.24	54.22 ± 1.15	54.68 ± 1.69	0.000 †	0.000 †	0.000 †	0.222
IMPA	95.06 ± 5.29	100.93 ± 4.71	96.95 ± 4.64	0.000 †	0.000 †	0.167	0.004 **
FMA	33.64 ± 5.68	26.43 ± 3.65	23.88 ± 3.93	0.000 †	0.000 †	0.000 †	0.045 *
U1-L1	113.05 ± 8.77	119.15 ± 8.36	123.32 ± 5.08	0.000 †	0.003 **	0.000 †	0.054
U1-SN	106.91 ± 7.61	104.02 ± 6.7	104.93 ± 4.28	0.228	0.093	0.270	0.612

* *p* < 0.05; ** *p* < 0.01; † *p* < 0.001.

**Table 3 diagnostics-13-00039-t003:** Comparison of alveolar bone morphology of central incisors between three groups.

	Group 1 (Class II; OB)	Group 2 (Class II; NOB)	Group 3 (Class I; NOB)	*p* (Group)	Comparison Between Groups		
					1 vs. 2	1 vs. 3	2 vs. 3
A1-height	15.35 (13.53~17.81)	15.73 (14.57~16.83)	15.65 (14.15~18.06)	0.747	/	/	/
A1-thickness	8.05 ± 1.29	9.25 ± 1.11	8.96 ± 1.01	0.000 †	0.000 †	0.005 **	0.355
A1-UBH	2.33 ± 0.51	2.07 ± 0.41	2.00 ± 0.41	0.020 *	0.031 *	0.009 **	0.574
A1-LBH	1.71 (1.40~2.03)	1.73 (1.50~1.95)	1.34 (1.21~1.52)	0.000 †	1.000	0.002 **	0.001 **
A1-OH	6.23 ± 2.14	5.27 ± 1.60	6.64 ± 1.97	0.242	0.331	0.451	0.095
C1-height	12.51 (10.85~14.77)	14.77 (12.69~17.11)	14.59 (13.07~15.75)	0.038 *	0.059	0.123	1.000
C1-thickness	6.95 ± 1.15	8.28 ± 1.21	8.58 ± 1.33	0.000 †	0.000 †	0.000 †	0.371
C1-UBH	2.59 ± 0.50	2.42 ± 0.61	2.20 ± 0.39	0.027 *	0.205	0.007 **	0.129
C1-LBH	2.33 ± 0.56	2.60 ± 0.50	2.35 ± 0.42	0.086	0.110	0.990	0.174
C1-OH	5.54 ± 2.65	6.07 ± 2.22	5.69 ± 2.25	0.680	0.393	0.811	0.565

* *p* < 0.05; ** *p* < 0.01; † *p* < 0.001.

**Table 4 diagnostics-13-00039-t004:** Comparison of alveolar bone morphological measurements between three groups.

	Group 1 (Class II; OB)	Group 2 (Class II; NOB)	Group 3 (Class I; NOB)	*p* (Group)	Comparison Between Groups		
					1 vs. 2	1 vs. 3	2 vs. 3
A3-height	13.47 ± 2.62	14.57 ± 1.84	14.94 ± 2.06	0.043 *	0.061	0.018 *	0.547
A5-height	12.49 (10.05~14.45)	11.25 (9.29~13.3)	10.85 (9.31~16.18)	0.320	/	/	/
A6-height	9.46 (8.02~12.5)	7.9 (6.85~9.07)	7.4 (5.84~10.56)	0.021 *	0.049 *	0.052	1.000
A7-height	10.59 ± 2.62	9.93 ± 2.34	9.26 ± 2.58	0.165	/	/	/
C3- height	12.08 (10.74~14.56)	14.32 (12.97~17.32)	13.96 (12.13~15)	0.012 *	0.010 *	0.274	0.796
C5-height	16.27 ± 3	16.84 ± 2.87	17.01 ± 1.98	0.569	/	/	/
C6M-height	15.36 ± 2.92	16.89 ± 2.39	17.01 ± 2.04	0.028 *	0.023 *	0.020 *	0.864
C6D-height	15.2 ± 2.94	16.63 ± 2.31	16.57 ± 1.78	0.050	0.072	0.109	0.996
C6-height	15.29 ± 2.86	16.76 ± 2.32	16.79 ± 1.86	0.033 *	0.023 *	0.027 *	0.963
C7M-height	13.43 (12.76~14.58)	14.98 (13.85~16.23)	14.32 (12.98~15.57)	0.038 *	0.031 *	0.738	0.605
C7D-height	13.43 ± 2.47	14.78 ± 2.25	14.1 ± 1.66	0.069	/	/	/
C7-height	13.44 (12.06~14.68)	15.24 (13.67~16.36)	14.52 (12.98~15.2)	0.017 *	0.013 *	0.418	0.649
A3-thickness	8.95 ± 1.16	10.89 ± 1.87	10.44 ± 1.26	0.000 †	0.000 †	0.000 †	0.279
A5-thickness	10.83 ± 1.62	12.18 ± 1.27	12.22 ± 1.74	0.001 **	0.001 **	0.002 **	0.922
A6-thickness	14.75 ± 1.56	15.98 ± 1.18	15.18 ± 0.93	0.002 **	0.000†	0.223	0.024 *
A7-thickness	15.37 (14.82~16.95)	16.8 (15.74~17.87)	15.82 (15.55~16.57)	0.033 *	0.042 *	1.000	0.148
C3-thickness	8.54 (7.96~9.4)	10.12 (9.41~11.65)	10.17 (9.49~11.1)	0.000 †	0.000 †	0.000 †	1.000
C5-thickness	10.29 (9.64~11.59)	10.86 (10.56~12.23)	11.52 (10.76~12.79)	0.003 **	0.054	0.003 **	0.910
C6M-thickness	11.85 (10.86~12.97)	12.94 (11.91~14.03)	13.11 (12.34~13.93)	0.001 **	0.007 **	0.002 **	1.000
C6D-thickness	12.33 ± 1.2	13.67 ± 1.38	13.72 ± 1.07	0.000 †	0.000 †	0.000 †	0.872
C7M-thickness	13.27 (12.46~14.54)	14.79 (13.65~15.66)	14.96 (13.59~15.46)	0.003 **	0.006 **	0.020 *	1.000
C7D-thickness	13.54 (12.79~15.23)	14.69 (13.92~15.83)	14.86 (13.69~15.48)	0.012 *	0.014 *	0.106	1.000

* *p* < 0.05; ** *p* < 0.01; † *p* < 0.001.

**Table 5 diagnostics-13-00039-t005:** Correlations between overbite and dentoalveolar parameters.

Parameter	Group 1 (Class II; OB)		Group 2 (Class II; NOB)		Group 3 (Class I; NOB)	
	r	*p*	r	*p*	r	*p*
U1-L1	−0.045	0.815	0.420 *	0.023	0.324	0.123
A1-OH	0.537 **	0.003	−0.131	0.499	0.030	0.888
C1-OH	0.286	0.133	0.317	0.094	0.146	0.496
A1-thickness	0.432 *	0.019	0.296	0.119	0.389	0.061
A3-thickness	0.011	0.954	0.450 *	0.014	0.347	0.097
A5-thickness	0.048	0.807	0.490 **	0.007	0.556 **	0.005
A6-thickness	0.015	0.940	0.229	0.233	0.323	0.124
A7-thickness	−0.270	0.157	0.187	0.331	0.294	0.163
C1-thickness	−0.346	0.066	0.496 **	0.006	0.518 **	0.009
C3-thickness	−0.264	0.166	0.149	0.440	0.364	0.081
C5-thickness	0.195	0.310	0.456 *	0.013	0.428 *	0.037
C6M-thickness	0.072	0.712	0.240	0.210	0.410 *	0.046
C6D-thickness	0.089	0.647	0.211	0.272	0.407 *	0.049
C7M-thickness	−0.035	0.859	0.047	0.808	0.310	0.140
C7D-thickness	−0.082	0.673	−0.054	0.780	0.371	0.074
A1-height	0.541 **	0.002	−0.113	0.561	0.308	0.143
A3-height	0.379 *	0.043	0.261	0.172	0.179	0.403
A5-height	−0.168	0.383	0.167	0.386	−0.073	0.736
A6-height	−0.521 **	0.004	0.295	0.121	−0.039	0.856
A7-height	−0.538 **	0.003	0.223	0.245	−0.110	0.608
C1-height	0.204	0.288	0.502 **	0.005	0.323	0.124
C3-height	0.188	0.329	0.119	0.538	0.180	0.399
C5-height	0.024	0.901	0.056	0.774	0.311	0.139
C6-height	0.149	0.441	−0.097	0.617	0.126	0.557
C7-height	0.158	0.414	−0.009	0.965	0.273	0.197

* *p* < 0.05; ** *p* < 0.01

## Data Availability

The data that support the findings of this study will be shared upon reasonable request to the corresponding author.

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
