# Peer review of "Three-Dimensional Analysis of Alveolar Bone Morphological Characteristics in Skeletal Class II Open Bite Malocclusion: A Cone-Beam Computed Tomography Study"

_diagnostics, 2022, doi:10.3390/diagnostics13010039_

Round 1

Reviewer 1 Report

The paper that studies the alveolar bone morphological characteristics in skeletal Class â…¡ open bite malocclusion by means of CT cone bean is more accurate if you compared it with the previous studies curried out using lateral ceph.

The article is well structured and clear, i think the English should be improved as the punctuation.

Further in the discussion you should add what are the clinical implications of the findings of this article. How then the class II OB patients should be treated (orthodontic treatment for example) taking into consideration what you found.

Author Response

Q1. The paper that studies the alveolar bone morphological characteristics in skeletal Class â…¡ open bite malocclusion by means of CT cone beam is more accurate if you compared it with the previous studies carried out using lateral ceph.

Response 1: We appreciate the reviewer’s constructive comments and suggestions. As mentioned in the introduction section, most of the previous literature has been measured with lateral cephalograms, but this two-dimensional image can only be evaluated from a single dimension, and their accuracy is susceptible to many factors, so conventional lateral cephalograms were not suitable for assessing alveolar bone morphology (Supontep Teerakanok et al.,2022). In addition, the two-dimensional image has the problem of magnification distortion, which does not accurately reflect the relationship between the posterior teeth and the maxillary sinus and mandibular canal (Sghaireen et al.2020), nor can it reflect the thickness of the alveolar bone (Asli Baysala et al.,2012; Xinnong Hu et al.2022). Therefore, many studies have shown that CBCT is an important tool for evaluating alveolar bone, and the comparison of the results of this experiment with previous studies has been discussed in the discussion section.

Q2: The article is well structured and clear; I think the English should be improved as the punctuation.

Response 2: Thank you for your comments! I have learned the correct usage of English punctuation and made improvements to the punctuation in this article.

Q3: Further in the discussion you should add what are the clinical implications of the findings of this article. How then the class II OB patients should be treated (orthodontic treatment for example) taking into consideration what you found.

Response 3: Thanks for your suggestion! The clinical significance of this experiment has been described separately in the discussion of the manuscript, and I have added a summary of clinical significance at the end of the article.

“Combined with the findings of this experiment, we recommend correcting Class II open bite malocclusion by appropriate maxillary first molars intrusion and avoiding extrusion of the incisors. At the same time, clinicians need to pay close attention to the position of the anterior teeth to ensure that the teeth are located in the alveolar bone to avoid the occurrence of dehiscence or fenestration.”

Reviewer 2 Report

Dear Authors, the study is detailed in parameteres but I didn't find information about race differences. Please refer this issue in a discussion. Did you find any relation to the age?

Author Response

Q1: Dear Authors, the study is detailed in parameters, but I didn't find information about race differences. Please refer this issue in a discussion.

Response 1: We appreciate your concerns and comments! We have reviewed the relevant information and have made additions to the article, which are as follows.

“Influenced by genetic factors, the incidence of anterior open bite varies greatly among different ethnicities, with the prevalence among the Caucasian population in America being 2.9% [2]. However, more comparative studies are needed to determine whether there are ethnic differences in the alveolar bone morphology of open bite malocclusion.”

Q2: Did you find any relation to age?

Response 2: We have done a Spearman correlation analysis between alveolar bone morphological indicators and age, and the results showed that the age of Class II open bite group was negatively correlated with the maxillary molar alveolar bone height and thickness. However, considering that the age span of the subjects in this experiment is not large, we believe that more rigorous experiments are needed to explore the relationship between age and alveolar bone morphology.

group 1

group 2

group 3

r

P

r

p

r

p

A1-thickness

0.082

0.671

-0.110

0.569

-0.027

0.901

A3-thickness

0.066

0.733

-0.029

0.880

-0.160

0.456

A5-thickness

-0.160

0.407

-0.270

0.157

-.459*

0.024

A6-thickness

-.507**

0.005

-0.310

0.101

-0.332

0.113

A7-thickness

-.453*

0.014

-0.321

0.089

-0.116

0.589

C1-thickness

-0.358

0.056

-0.249

0.193

0.136

0.525

C3-thickness

-0.338

0.073

-0.080

0.678

0.081

0.707

C5-thickness

0.017

0.928

0.286

0.133

-0.179

0.402

C6M-thickness

0.015

0.940

.385*

0.039

-0.168

0.432

C6D-thickness

0.016

0.933

0.327

0.083

-0.085

0.693

C7M-thickness

0.107

0.580

0.241

0.207

-0.064

0.765

C7D-thickness

0.101

0.602

0.016

0.935

-0.029

0.893

A1-height

0.215

0.263

0.202

0.293

-0.016

0.940

A3-height

.478**

0.009

0.010

0.958

0.093

0.667

A5-height

-0.093

0.632

0.149

0.442

0.066

0.761

A6-height

-.379*

0.043

0.012

0.952

0.156

0.466

A7-height

-.430*

0.020

-0.141

0.465

-0.073

0.734

C1-height

0.036

0.851

-.434*

0.019

-0.082

0.704

C3-height

-0.102

0.600

-0.078

0.689

-0.222

0.296

C5-height

-0.206

0.283

0.020

0.918

-0.095

0.659

C6-height

-0.360

0.055

0.249

0.192

-0.090

0.675

C7-height

-.478**

0.009

0.354

0.059

-0.040

0.853